# Administration of nucleoside-modified mRNA encoding broadly neutralizing antibody protects humanized mice from HIV-1 challenge

Norbert Pardi[1], Anthony J. Secreto[1], Xiaochuan Shan[1], Fotini Debonera[1], Joshua Glover[1], Yanjie Yi[1], Hiromi Muramatsu[1], Houping Ni[1], Barbara L. Mui[2], Ying K. Tam[2], Farida Shaheen[1], Ronald G. Collman[1], Katalin Karikó[1], Gwenn A. Danet-Desnoyers[1], Thomas D. Madden[2], Michael J. Hope[2] & Drew Weissman[1]

Monoclonal antibodies are one of the fastest growing classes of pharmaceutical products, however, their potential is limited by the high cost of development and manufacturing. Here we present a safe and cost-effective platform for *in vivo* expression of therapeutic antibodies using nucleoside-modified mRNA. To demonstrate feasibility and protective efficacy, nucleoside-modified mRNAs encoding the light and heavy chains of the broadly neutralizing anti-HIV-1 antibody VRC01 are generated and encapsulated into lipid nanoparticles. Systemic administration of $1.4\,mg\,kg^{-1}$ of mRNA into mice results in $\sim170\,\mu g\,ml^{-1}$ VRC01 antibody concentrations in the plasma 24 h post injection. Weekly injections of $1\,mg\,kg^{-1}$ of mRNA into immunodeficient mice maintain trough VRC01 levels above $40\,\mu g\,ml^{-1}$. Most importantly, the translated antibody from a single injection of VRC01 mRNA protects humanized mice from intravenous HIV-1 challenge, demonstrating that nucleoside-modified mRNA represents a viable delivery platform for passive immunotherapy against HIV-1 with expansion to a variety of diseases.

[1] Department of Medicine, University of Pennsylvania, Philadelphia, Pennsylvania 19104, USA. [2] Acuitas Therapeutics, Vancouver, British Columbia, Canada V6T 1Z3. Correspondence and requests for materials should be addressed to D.W. (email: dreww@mail.med.upenn.edu).

Monoclonal antibodies have emerged as highly effective therapeutics for cancer, infectious diseases and auto-immune disorders (reviewed in ref. 1). However, the high cost of recombinant protein production limits its use worldwide, but particularly in resource-limited settings. Maintaining effective levels of therapeutic antibody (Ab) requires frequent injection based on clearance and therapeutic goal[2]. Vector-mediated antibody gene transfer overcomes these challenges by enabling high levels of Ab production *in situ* for extended durations. At present, the preferred vector is recombinant adeno-associated virus (AAV) containing the Ab gene of interest[3]. AAV is a single stranded DNA virus, which exists in an extra chromosomal state after infection. A recent finding of AAV2 integration into known cancer genes in hepatocellular carcinomas raises concerns about the safety of this therapeutic approach[4]. Another problem associated with AAV is its immunogenicity. Despite optimization, AAV remains immunogenic and AAV-mediated gene transfer of Ab, in certain instances, induced blocking anti-idiotype Ab[5]. *In vitro*-transcribed (IVT) mRNA, the minimal vector encoding genetic information, has fundamental advantages over DNA- and viral-based systems, such as only requiring direct uptake into the cytosol, which results in immediate protein production, and the lack of genomic integration. This makes mRNA a safe and fully controllable delivery tool. Moreover, unlike protein-based therapeutics, production of mRNA is simple and cost effective; high levels of therapeutic protein are produced, folded and modified by host cells and the delivered mRNA is continuously translated for extended and controllable durations.

To make IVT mRNA suitable for therapy, several qualities, including stability and translatability have been improved. We previously demonstrated that optimized coding sequence, selected 3′- and 5′-UTRs, enzymatically generated 5′ cap1 structures and long 3′ poly(A)-tail render IVT mRNA substantially more stable and translatable. Moreover, FPLC purification and incorporation of modified nucleosides, such as 1-methylpseudouridine (m1Ψ), made the mRNA non-immunogenic and significantly increased translation[6–8]. Formulation and route of delivery *in vivo* also impact the translational kinetics of the IVT mRNA. Using a mouse model, we recently demonstrated that lipid nanoparticles (LNPs) are efficient mRNA carriers enabling high levels of protein production for extended periods of time when administered by a variety of routes[9].

Several studies have reported that RNA-based vaccines could elicit strong antigen-specific T and B cell immune responses against infectious pathogens[10–15]. The inherent adjuvant activity of RNA might be beneficial for vaccination but is detrimental for applications in which the mRNA encodes a therapeutic protein. For this, the IVT mRNA needs to be non-immunogenic to avoid adverse events, including the release of proinflammatory cytokines, the inhibition of translation and the generation of anti-drug antibodies.

Here we demonstrate that nucleoside-modified mRNA encapsulated into LNPs is an effective tool for protein therapy. Using LNP-formulated, m1Ψ-containing mRNAs encoding the light and heavy chains of VRC01, a broadly neutralizing antibody against HIV-1 (ref. 16), we demonstrate that systemically delivered mRNA-LNPs are quickly translated into functional antibody. Furthermore, we show that a single injection of VRC01 mRNA-LNPs can fully protect humanized mice against intravenous challenges with the SF162 and JR-CSF HIV-1 isolates. These findings serve as the basis for the use of the nucleoside-modified mRNA-LNP platform for delivery of anti-HIV-1 Abs, as well as, other therapeutic antibodies and proteins.

## Results

**Administration of a single dose of VRC01 mRNA-LNPs.** In accordance with our previous findings, intravenous (i.v.) delivery of mRNA-LNPs resulted in robust protein expression in the liver (Supplementary Fig. 1 and ref. 9). To determine the kinetics of VRC01, a human monoclonal antibody, production from mRNA-LNPs, BALB/c mice were injected i.v. with a single dose of 30 μg (1.4 mg kg$^{-1}$) of LNP-formulated m1Ψ-modified mRNA encoding the heavy and light chains of the VRC01 Ab in equimolar concentrations or the control firefly luciferase (Luc). VRC01 levels were measured every other day for 11 days (Fig. 1). Antibody levels peaked at 24 h post injection and displayed a gradual decrease, remaining between 130–170 μg ml$^{-1}$ for 5 days. A sharper decrease in the level of VRC01 Ab was observed by day 7 and antibody levels were below detection at day 11 post injection. The very small error bars indicate that similar levels of VRC01 Ab were measured in each animal injected, demonstrating the reproducibility of dosing with nucleoside-modified mRNA-LNPs. The increased rate of clearance of human IgG in mice is noted.

**Weekly injections of VRC01 mRNA-LNPs.** To avoid antibody induction against the VRC01 human protein in mice, NOD-scid gamma (NSG) mice were used. Animals received five i.v. injections of 30 μg (1 mg kg$^{-1}$) of VRC01-encoding m1Ψ-modified mRNA-LNPs at a weekly interval and antibody levels were measured 7 days after each injection (Fig. 2). High VRC01 Ab levels could be maintained by weekly injections. An important observation from this data was that there was no reduction in the level of translation with subsequent injections. Antibody levels maintained a trough level between ∼40 and 60 μg ml$^{-1}$.

**Investigation of immune activation by VRC01 mRNA-LNPs.** It is well documented that systemic delivery of nucleic acids, including conventional and unpurified IVT mRNA, induces immune activation that results in production of type I interferons (IFNs) and proinflammatory cytokines[8,17]. To determine immune activation by the nucleoside-modified, FPLC-purified mRNA-LNPs, C57Bl/6 mice were injected i.v. with a single dose (1 mg kg$^{-1}$) of LNP-formulated mRNA encoding firefly luciferase (Luc). Plasma was collected at 2 and 4 h post injection and type I IFN and proinflammatory cytokine levels were measured by Luminex assay (Fig. 3). We have previously found that unpurified nucleoside-modified mRNA always induces a low but measurable level of immune activation and FPLC

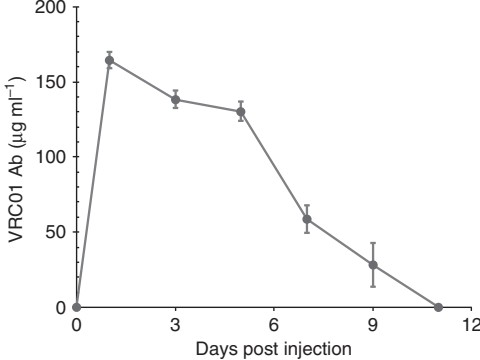

**Figure 1 | Kinetics of VRC01 production after a single injection of mRNA-LNPs.** BALB/c mice were i.v. injected with 30 μg (1.4 mg kg$^{-1}$) of VRC01-encoding m1Ψ-mRNA-LNPs. Animals were bled every other day and VRC01 levels in serum were measured by ELISA. Error bars are standard error of the mean. Group size is 6 animals.

purification removes all immunogenicity[8]. Thus, a commercially available, Luc-encoding, nucleoside-modified, unpurified (immunogenic) mRNA was included in the study as a positive control. No increase in levels of interleukin-6, IFN-α and tumour necrosis factor-α was measured in plasma obtained from mice that were injected with LNP-formulated, FPLC-purified, m1Ψ mRNA-LNPs.

To demonstrate that nucleoside-modified mRNA administration does not induce antibody production against the encoded protein, BALB/c mice received weekly intraperitoneal injections of 0.1 μg of murine erythropoietin (muEPO) encoding mRNA and anti-EPO antibody levels from mouse plasma were monitored over time. No anti-EPO Ab production was detected after 5 weekly injections (Supplementary Fig. 2).

**Protection from HIV-1 challenge by mRNA-LNPs.** Prophylactic administration of potent monoclonal antibodies against infectious pathogens can protect healthy individuals from infection (reviewed in ref. 18). To investigate whether systemic delivery of mRNA-LNPs encoding VRC01 could protect animals from HIV-1 viral challenge, two forms of humanized mice were used. CD34-NSG humanized mice were generated by injection of human CD34$^+$ stem cells. This model system has circulating and splenic human CD4$^+$ T cells but no human lymphoid cells in tissues, including genital tract, lymph nodes, and the gastrointestinal tract[19], therefore they can be infected with HIV-1 after parenteral virus administration. The second model

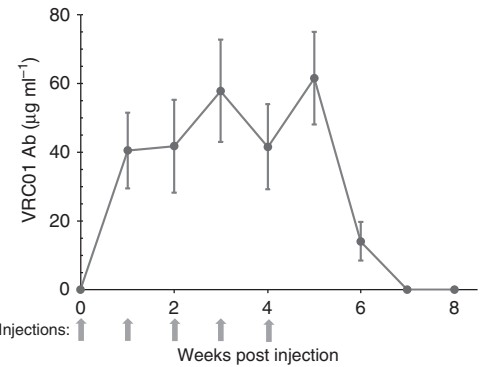

**Figure 2 | Kinetics of VRC01 production after weekly injections of mRNA-LNPs.** NSG mice were i.v. injected weekly with 30 μg (1 mg kg$^{-1}$) of VRC01 m1Ψ-mRNA-LNPs. Animals were bled prior to each injection (7 days after the preceeding injection) and VRC01 levels were measured by ELISA. Grey arrows indicate mRNA-LNP injections. Error bars are s.e.m. Group size is three animals.

were BLT (bone marrow, liver, thymus) mice that also contain lymphoid human cells[20]. Typically, approximately half of the lymphoid cells in the blood are of human origin (Supplementary Figs 3a and 4). Animals had normal T and B cell ratios and CD4$^+$ and CD8$^+$ T cell ratios (Supplementary Fig. 3b,c).

A dose response experiment was performed to characterize antibody production from various amounts of VRC01 mRNA-LNPs in BLT humanized mice. Moreover, to directly compare the efficacy of the nucleoside-modified mRNA-LNP platform to recombinant protein delivery, a group of animals was injected with recombinant, purified VRC01 mAb. A single dose of 30 μg (1.4 mg kg$^{-1}$), 15 μg (0.7 mg kg$^{-1}$) or 7.5 μg (0.35 mg kg$^{-1}$) of m1Ψ-modified VRC01 mRNA-LNPs or 600 μg (28 mg kg$^{-1}$) of VRC01 mAb or 30 μg of Luc-encoding mRNA-LNPs was administered by the i.v. route and VRC01 Ab concentrations were measured 24 h post-injection. Strikingly, plasma VRC01 levels were 1.65 times higher in mice injected with 30 μg of VRC01 mRNA-LNPs compared to animals injected with 600 μg of VRC01 mAb (mean values of 204.7 μg ml$^{-1}$ and 123.4 μg ml$^{-1}$, respectively) (Fig. 4a). A mean plasma VRC01 concentration of 65.5 μg ml$^{-1}$ and 23.5 μg ml$^{-1}$ was measured in animals injected with the 15 μg and 7.5 μg VRC01 mRNA-LNP doses, respectively. After confirmation of high antibody levels in the VRC01 injected animals, mice were challenged with the SF162 HIV-1 isolate by the i.v. route. Mice were bled 2 weeks post-challenge and viral RNA levels were measured in plasma by quantitative real-time PCR (qRT–PCR; Fig. 4b). Viral RNA was detected in all Luc mRNA-injected animals and 5 of 6 animals that received the lowest dose (7.5 μg) of VRC01 mRNA-LNPs. The latter data demonstrates that 23.5 μg ml$^{-1}$ VRC01 Ab concentration in the plasma is not protective against challenge with the SF162 HIV-1 isolate. Viral loads were below detection (1 copy per μl plasma, 10 μl was used for the measurement) in all mice that were treated with 30 μg or 15 μg of VRC01 mRNA-LNPs or 600 μg VRC01 protein before HIV-1 challenge.

To confirm our findings, HIV-1 challenge experiments were repeated with the JR-CSF HIV-1 viral isolate. The IC$_{50}$'s of neutralization of JR-CSF and SF162 are 0.164 and 0.421 μg ml$^{-1}$, respectively, demonstrating that SF162 is approximately three times more difficult to neutralize[21]. CD34-NSG humanized mice were injected i.v. with 30 μg (1 mg kg$^{-1}$) of VRC01 or Luc mRNA-LNPs and challenged with virus 24 h later after confirmation of high VRC01 Ab levels in the VRC01 mRNA-LNP-injected animals (Fig. 5a). High HIV-1 RNA levels were detected in all Luc mRNA-injected animals and no virus was detectable in the VRC01 mRNA-LNP-injected mice 1 and 2 weeks post-challenge (Fig. 5b). Broadly neutralizing HIV-1 monoclonal antibodies can suppress viral replication or prevent infection[22]. To determine whether administration of VRC01

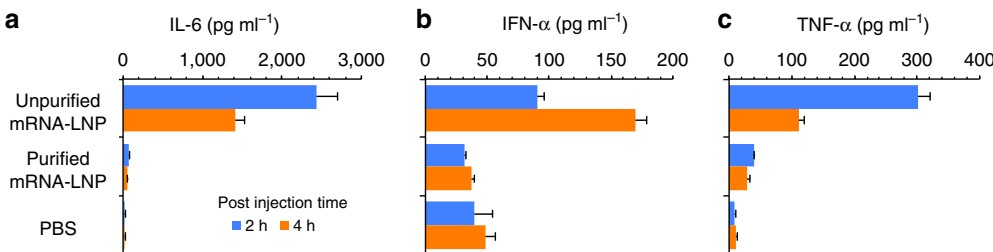

**Figure 3 | Analysis of innate immune activation by mRNA-LNPs.** C57Bl/6 mice were i.v. injected with a 1 mg kg$^{-1}$ dose of nucleoside-modified, FPLC-purified Luc mRNA-LNPs; unpurified, nucleoside-modified Luc mRNA-LNPs (1 mg kg$^{-1}$) (positive control) and phosphate buffered saline (PBS) (negative control). Animals were bled 2 and 4 h post injection and interleukin-6 (**a**), IFN-α (**b**) and tumour necrosis factor-α (**c**) levels were measured in plasma by Luminex assay. Error bars are s.e.m. Statistical analysis: one-way analysis of variance with Bonferroni correction, $P < 0.01$ in comparisons of PBS to non-purified mRNA-LNPs and non-purified mRNA-LNPs to purified mRNA-LNPs. Group size is five animals.

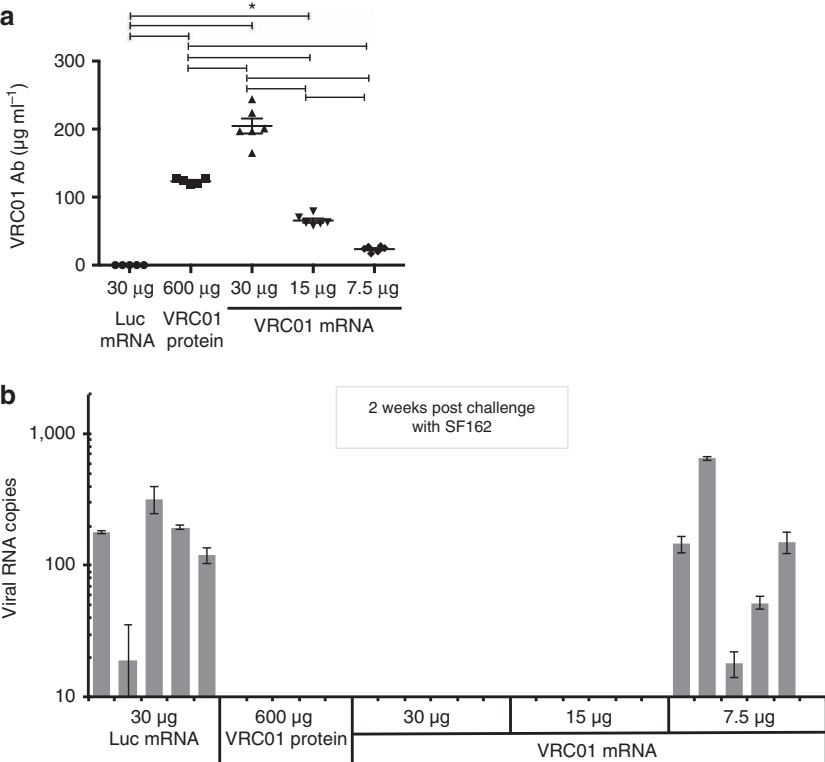

**Figure 4 | SF162 HIV-1 challenge of humanized mice treated with VRC01 mRNA-LNPs or VRC01 protein.** BLT humanized mice were i.v. injected with 30 µg of Luc mRNA-LNP (negative control), 600 µg (28 mg kg$^{-1}$) VRC01 mAb or 30 µg (1.4 mg kg$^{-1}$), 15 µg (0.7 mg kg$^{-1}$), 7.5 µg (0.35 mg kg$^{-1}$) of VRC01 mRNA-LNP. (**a**) Animals were bled 24 h post-injection and VRC01 mAb levels were measured in the plasma by ELISA. (**b**) Viral loads (RNA copies per 10 µl plasma) at 2 weeks post-challenge with the SF162 HIV-1 isolate were measured by qRT–PCR. Error bars are s.e.m. Statistics: (**a**) One-way analysis of variance with Bonferroni corrections $P < 0.001$ in all marked categories, (**b**) Student's $t$-test, two-sided, comparing Luc treated to VRC01 treated were highly significant: $P < 0.001$. Group size is five or six animals.

mRNA-LNPs fully protected humanized mice from HIV-1 infection, integrated HIV-1 gag DNA was measured by performing conventional PCR on genomic DNA isolated from spleen at 3 weeks post JR-CSF HIV-1 challenge. Integrated gag DNA was detected in all Luc mRNA-injected animals (Fig. 5c). No measurable levels of HIV-1 gag DNA were detected in mice injected with VRC01 mRNA-LNPs prior to infection.

## Discussion

Monoclonal antibody therapeutics are among the fastest growing segments of pharmaceutical development with applications in cancer, autoimmune diseases and bone metabolism (reviewed in ref. 1). A great body of knowledge has been accumulated on the administration of Abs as proteins or encoded by viruses and DNAs, but there is no available data on the investigation of Ab-encoding mRNA therapy. In this report, we evaluated the kinetics and protective efficacy of nucleoside-modified, FPLC-purified mRNA-based delivery of the VRC01 broadly neutralizing anti-HIV Ab.

Therapeutic monoclonal antibodies are typically produced by cell lines, such as Chinese hamster ovary cells, followed by extensive purification of the antibody from cell culture supernatant and pharmaceutical formulation. Production of antibodies has many challenges, including misfolding or incorrect post-translational modification that can lead to adverse events. The purification is specific for each type of monoclonal antibody, requiring the development of a new method for each, thus making production expensive. In contrast, production and purification of IVT mRNA is simple, fast and cost effective, as it does not require complex and expensive laboratory infrastructure and the same methods can be used for all mRNAs[23,24].

Transfer of genes encoding antibodies is a novel strategy and the most common gene transfer vehicle is AAV (reviewed in ref. 3), although lentiviral vectors and plasmids have also been studied. Muthumani *et al.*[25] electroporated 25 µg of VRC01 Fab-encoding plasmid, intramuscularly. This approach yielded modest levels (2–3 µg ml$^{-1}$) of VRC01 Fab in serum measurable for over 10 days post administration. Gene transfer of anti-HIV Ab by lentiviral vectors enabled stable Ab production, however the need for *ex vivo* transduction of cells with the vector greatly limits the use of this platform in large scale studies[26,27]. Balazs *et al.*[28] performed AAV vector-mediated gene transfer studies where they expressed native anti-HIV antibodies, including VRC01 in mice. Depending on the administered dose, antibody concentrations could peak higher than 100 µg ml$^{-1}$ after intramuscular injection of AAV and lasted over a year. Protection from HIV-1 viral challenge in humanized mice was also demonstrated. The use of AAV vector-mediated antibody gene transfer has been extended to other targets, such as RSV, influenza and cancer (reviewed in ref. 3). However, the identification of AAV genomes in tumour-associated genes of hepatocellular carcinomas has recently raised safety concerns about the therapeutic use of AAVs[4]. In addition, the immunogenicity of AAVs has been associated with clearance of virally infected cells resulting in the loss of protein expression and the induction of immune responses against the encoded protein,

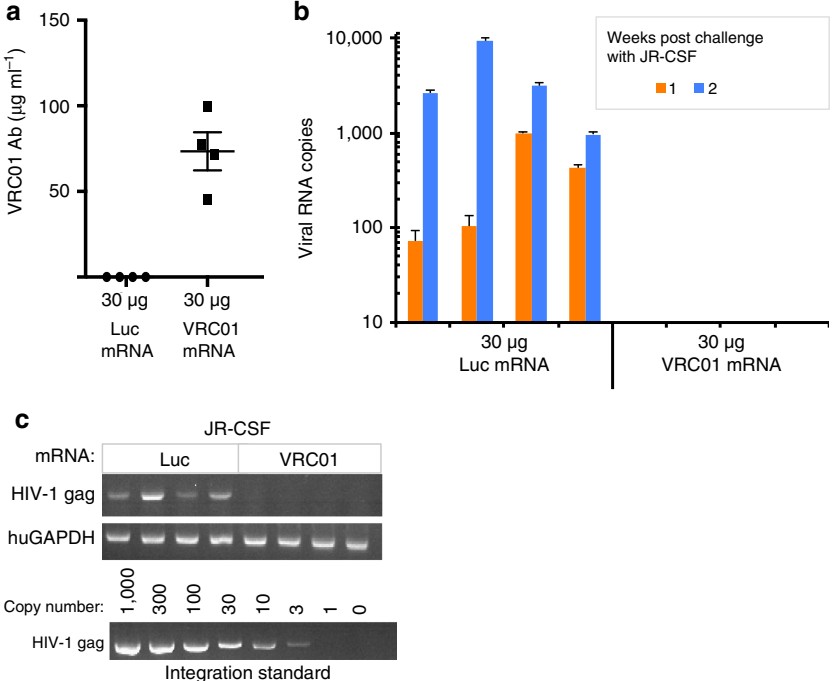

**Figure 5 | JR-CSF HIV-1 challenge of humanized mice treated with VRC01 mRNA-LNPs.** CD34-NSG humanized mice were i.v. injected with 30 μg (1 mg kg$^{-1}$) of VRC01 or 30 μg of Luc mRNA-LNPs. (**a**) Animals were bled 24 h post-injection and VRC01 levels were measured in the plasma by ELISA. (**b**) Viral loads (RNA copies per 10 μl plasma) at 1 and 2 weeks post-challenge were measured by qRT-PCR. (**c**) Gag DNA integration was measured in splenocytes by conventional PCR 3 weeks post challenge. Ultraviolet-illuminated ethidium bromide-stained PCR products are shown. The pNL4-3 integration standard was included to determine the sensitivity of the PCR assay. Error bars are s.e.m. Student's t-test, two-sided, comparing Luc treated to VRC01 treated were highly significant: (**a**) $P < 0.001$, (**b**) $P < 0.001$. Group size is four animals.

even when it was endogenously expressed, such as erythropoietin[29]. As VRC01 is a human protein, we could not evaluate its immunogenicity. In previous studies, where erythropoietin encoding mRNA was repeatedly given to mice at a weekly interval, no antibody responses were found[17] (Supplementary Fig. 2). IVT mRNA holds promise to be a safe and controllable alternative to DNA and viral based therapeutic delivery platforms. The discoveries that nucleoside modification and FPLC purification of IVT mRNA makes it unable to activate RNA sensors and induce type I IFNs, proinflammatory cytokines, and translation suppressive effectors (reviewed in refs 30,31) opens up an array of therapeutic applications, including protein replacement, gene therapy, and protein therapeutics. A series of studies have recently been published using modified nucleoside-containing IVT mRNA for protein therapy in mice and monkeys demonstrating its feasibility for development of human therapeutics[17,32–36]. Moreover, a great variety of mRNA complexing formulations have been developed, and lipid nanoparticles, originally created for siRNA delivery[37–40], have proven to be safe and very efficient mRNA carriers for *in vivo* administration[9].

A number of important features of nucleoside-modified mRNA delivered in LNPs are demonstrated. First, a single i.v. injection of 30 μg (1.4 mg kg$^{-1}$) of VRC01 mRNA-LNPs into BALB/c mice consistently resulted in serum antibody concentrations of 170 μg ml$^{-1}$ at 24 h post injection across treated mice, which greatly exceeds the half maximal inhibitory concentration (IC$_{50}$) of VRC01 Ab against a variety of HIV-1 isolates[21,41]. High antibody concentrations could be measured for up to 9 days and could be maintained by repeated weekly injections. Second, the inability of nucleoside-modified and FPLC-purified mRNA-LNPs to activate the innate immune system was

demonstrated in 2 manners. The clinically most important was the demonstration that repeated deliveries at one week intervals resulted in no decrease in the levels of translation of subsequent doses. In addition, no increase in type I IFN and proinflammatory cytokine production was detected after i.v. administration of nucleoside-modified and FPLC-purified mRNA-LNPs (Fig. 3).

Prophylactic immunization with monoclonal antibody protein in humanized mice typically uses 10–20 mg kg$^{-1}$ doses to reach therapeutic concentrations[22,42]. In this study, we demonstrated that 15 μg (0.7 mg kg$^{-1}$) of nucleoside-modified mRNA encapsulated in LNPs delivered by the i.v. route protected humanized mice from HIV challenge. The data presented here demonstrate that the nucleoside-modified mRNA-LNP platform is a safe, simple, and efficient alternative to therapeutic protein delivery with potential application to any monoclonal antibody with extension to any therapeutic protein.

## Methods

**Ethics statement.** *Animals.* The investigators faithfully adhered to the 'Guide for the Care and Use of Laboratory Animals' by the Committee on Care of Laboratory Animal Resources Commission on Life Sciences, National Research Council. All animal facilities are fully accredited/certified by the American Association for Accreditation of Laboratory Animal Care (AAALAC) or Canadian Council on Animal Care (CCAC). All studies were conducted under protocols approved by the appropriate Institutional Animal Care and Use Committees.

**mRNA production.** mRNAs were produced as previously described[23] using T7 RNA polymerase (Megascript, Ambion) on linearized plasmids encoding codon-optimized firefly luciferase (pLuc19), codon-optimized mouse erythropoietin (pTEV-muEPO-A101) and light and heavy chains of VRC01 (pTEV-VRC01-LC-A101, pTEV-VRC01-HC-A101). mRNAs were transcribed to contain 110 nt (pLuc19) or 101 nt (pTEV-VRC01-LC-A101 and pTEV-VRC01-HC-A101) poly(A) tails. m1Ψ-5′-triphosphate (TriLink) instead of UTP was used to generate modified nucleoside-containing mRNA. RNAs were capped using the

m7G capping kit with 2′-O-methyltransferase (ScriptCap, CellScript) to obtain cap1. mRNA was purified by Fast Protein Liquid Chromatography (FPLC) (Akta Purifier, GE Healthcare), as described[24]. All RNAs were analysed by electrophoresis using denaturing or native agarose gels, and stored at −20 °C. Unpurified pseudouridine (Ψ)-modified luciferase-encoding mRNA was obtained from TriLink.

**Formulation of mRNA in lipid nanoparticles.** IVT mRNAs encoding the light and heavy chains of VRC01 were mixed in 1:1 molar ratio before LNP encapsulation. FPLC-purified m1Ψ-containing firefly luciferase or VRC01 light and heavy chain-encoding mRNAs were encapsulated in LNPs using a self-assembly process in which an aqueous solution of mRNA at pH 4.0 is rapidly mixed with a solution of lipids dissolved in ethanol[38]. LNPs used in this study were similar in composition to those described previously[37,38], which contain an ionizable cationic lipid/phosphatidylcholine/cholesterol/PEG-lipid (50:10:38.5:1.5 mol mol$^{-1}$) and were encapsulated at an RNA-to-total lipid ratio of ~0.05 (wt/wt). They had a diameter of ~80 nm, as measured by dynamic light scattering using a Zetasizer Nano ZS (Malvern Instruments Ltd, Malvern, UK) instrument. mRNA-LNP formulations were stored at −80 °C at a concentration of mRNA of ~1 µg µl$^{-1}$.

**Complexing of muEPO mRNA.** RNA was complexed to TransIT-mRNA (Mirus Bio, Madison, WI) according to the manufacturer's instructions. A ratio of mRNA (0.1 µg), TransIT-mRNA reagent (0.11 µl), and Boost reagent (0.07 µl) in a final volume of 10 µl Dulbecco's modified Eagle's medium was used. For complexing different amounts of mRNA, the volumes of the reagents and the final volume were scaled proportionally.

**Mice.** *BALB/c mice.* Female BALB/c mice were purchased from Harlan Laboratories. C57Bl/6 mice: animals were obtained from Charles River Laboratories.

*CD34-NSG mice.* NSG mice were obtained from the Jackson Laboratory (strain number 005557). Male and female mice between 6–8 weeks of age were used at the initiation of each experiment. Humanized CD34-NSG mice were generated by injecting NSG mice i.v. with 100,000 human CD34$^+$ hematopoietic stem cells after conditioning with busulfan (30 mg kg$^{-1}$) for 24 h. 12 weeks after stem cell injection, mice were bled and engraftment of human immune cells was confirmed by staining with the CD45-PE (BD Biosciences, Cat. Number 555483), CD19-PerCP-Cy5.5 (BD Biosciences, Cat. Number 340951), CD33-FITC (BD Biosciences, Cat. Number 555626) and CD3-APC (BD Biosciences, Cat. Number 555335) antibodies and performing flow cytometric analyses (Supplementary Fig. 4).

*BLT mice.* NSG mice were bred in house using original breeding stock obtained from the Jackson Laboratory (strain number 005557). Male and female mice between 7–10 weeks of age were used for human immune system reconstitution 12–16 weeks prior to experimentation. BLT mice are NSG mice with a reconstituted human immune system, generated by injecting NSG mice i.v. with 100–125,000 human CD34$^+$ hematopoietic stem cells 24 h post busulfan (30 mg kg$^{-1}$) conditioning. Mice were surgically implanted 3–6 days post stem cell transplant with 3–5 pieces of human fetal thymus measuring approximately 3–5 cubic mm under the renal capsule. Engraftment of human cells in BLT mice was assessed 12–16 weeks after stem cell injection, by staining with the CD45-PE (BD Biosciences, Cat. Number 555483), CD19-PerCP-Cy5.5 (BD Biosciences, Cat. Number 340951), CD33-APC (BD Biosciences, Cat. Number 551378), CD4-PerCp-Cy5.5 (BD Biosciences, Cat. Number 560650), CD8-FITC (BD Biosciences, Cat. Number 555366) and CD3-APC (BD Biosciences, Cat. Number 555335) antibodies and performing flow cytometric analyses (Supplementary Fig. 3).

Power analysis was used to calculate the size of all animal groups to ensure statistically significant results based on preliminary studies. Experiments were performed 1 or more times to achieve the needed number of mice in each group to achieve statistical significance. No randomization was used, beyond random selection of mice into different treatment groups. All analyses of treated mice were performed in the absence of knowledge of the treatment groups.

**Administration of VRC01 mAb and VRC01 mRNA-LNPs to mice.** All reagents were diluted in Dulbecco's Phosphate Buffered Saline (PBS) and injected into animals i.v. with a 3/10cc 29½G insulin syringe (BD Biosciences). Several independent mRNA syntheses and LNP encapsidations were used, but the same preparation was used in BALB/c, NSG and CD34-NSG mice, which resulted in different amounts of VRC01 circulating protein 24 h later.

**Administration of TransIT-complexed mRNA into mice.** The muEPO-encoding mRNA was injected into the peritoneal cavity with a 27-gauge needle using standard technique.

**Blood collection.** Blood was collected from the tail vein or from the peri-orbital venous sinus. Blood was centrifuged for 10 min at 3,000 r.p.m. and the plasma was stored at −80 °C until analyses.

**Enzyme-linked immunosorbent assays.** VRC01 antibody levels were determined by ELISA using the VRC01 monoclonal antibody (NIH AIDS Reagent Program, Cat. Number 120033) as a standard. Immulon 4 HBX high-binding plates were coated with 100 µl of purified HIV-1 HXBc2 gp120 at a final concentration of 1 µg ml$^{-1}$ in PBS overnight at 4 °C. The plates were washed four times with wash buffer (0.05% Tween-20 in PBS) and then blocked with blocking buffer (2% BSA in PBS) for 1 h at room temperature, after which the plates were washed three more times with wash buffer. Dilutions of plasma samples and standard were made in blocking buffer (2% BSA in PBS) and incubated (100 µl per well) for 1 h at room temperature. Samples and standard were removed and the plate was washed four times with wash buffer. Detection antibody (Bethyl Laboratories, Cat. Number A80-304 P) was diluted 1:100,000 in blocking buffer and incubated (100 µl per well) for 1 h. After four washes, TMB substrate mixture (KPL) was added at 100 µl per well for 20 min. 2 N sulfuric acid (50 µl per well) was used to stop the reaction, and the optical density was read at 450 nm on a Dynex MRX Revelation microplate reader.

Anti-muEPO Ab responses were measured by coating plates with muEPO protein (R&D systems, Cat. Number 959-ME), as described above. Plasma from mice receiving weekly muEPO encoding mRNA was diluted 1:10 in phosphate-buffered saline and added to muEPO protein coated plates. A positive control containing a rat anti-murine EPO mAb (R&D Systems) spiked into normal mouse plasma was used. Bound antibodies were detected with goat anti-mouse and rat Ig antisera labelled with peroxidase (Sigma). A concentration of 0.1 ng ml$^{-1}$ of anti-muEPO antibody spiked into plasma-phosphate-buffered saline could be detected.

**Luminex assay.** Blood was collected at 2 and 4 h post i.v. injection of mRNA-LNPs or PBS. Plasma was obtained and stored, as described above. Type I IFN, cytokine and chemokine levels were analysed using an Affymetrix (Santa Clara, CA) 36-analyte xMAP ProcartaPlex multiplex immunoassay on a Bio-Plex 200 (BioRad, Hercules, CA) device and data was evaluated using xPonent software (V3.1) following the manufacturer's instructions.

**Intravenous HIV-1 challenge of humanized mice.** Twenty four hours after intravenous delivery of VRC01 or Luc mRNA-LNPs or VRC01 protein, humanized mice were bled and VRC01 antibody levels were determined by ELISA. Mice were then challenged intravenously with JR-CSF (10 ng p24) or SF162 (50 ng p24) grown in PHA-blasts and obtained from the Penn CFAR core and diluted in PBS to a volume of 100 µl. Blood samples of infected mice were obtained weekly to determine viral load (RNA copy numbers) in plasma. Three weeks after infection, mice were killed and spleens dissected into single cell suspensions.

**Genomic DNA isolation from mouse spleen.** QIAamp DNA Micro Kit (Qiagen) was used to isolate genomic DNA from spleen, as described in the QIAamp DNA Micro Handbook, except that the final sample was dissolved in ultrapure water.

**Conventional polymerase chain reaction (PCR).** HIV-1 gag PCR was performed by using 50 ng of splenic genomic DNA and the G20 (5′-GTATGGGCAAGCAG GGAGCTAGAA-3′) and G55 (5′-ATTTCTCCCACTGGGATAGGTGG-3′) gag-specific primers in a 50 µl reaction volume. To determine the sensitivity of PCR, an HIV-1 integration standard (pNL4-3) was used as a template under the same conditions[43]. GAPDH PCR was carried out using the following primers: huGAPDH fw: 5′-CCACCCATGGCAAATTCCATGGCA-3′ and huGAPDH rev: 5′-TCTAGACGGCAGGTCAGGTCCACC-3′. The thermal cycler (Techne TC-5000, Bibby Scientific) was programmed to perform a 5-min hot start at 94 °C, followed by 40 cycles: denaturation at 94 °C for 1 min, annealing at 55 °C for 1 min and extension at 72 °C for 1 min. A 10-min final extension at 72 °C was performed. PCR products were analysed on a 2% agarose gel with 1 µg ml$^{-1}$ ethidium bromide (Sigma) and was recorded with a UV transilluminator (GelDoc 1000 gel imaging system, Bio-Rad).

**Plasma viral load measurement.** RNA was extracted directly from 30 µl of EDTA-plasma using the method of Cillo et al.[44] and the RNA pellet was reconstituted in a final volume of 15 µl. Before extraction, a uniform quantity of Replication Competent Avian Sarcoma (RCAS) virions were spiked into each sample and amplified separately to verify virus/RNA recovery and lack of PCR inhibition[45]. RNA was reverse transcribed using random hexamers and subjected to realtime qPCR quantification using the LightCycler 480 Probes Master (Roche; Indianapolis, IN), as described in Cillo et al.[44]. For each sample, the qRT–PCR reaction was run in duplicate on 5 µl RNA (equivalent to 10 µl plasma); no-reverse transcriptase reaction and RCAS amplification were run on one well per sample using 2.5 µl RNA. The HIV-1 primer/probe targets the pol gene and detects all group M clades (forward: 5′-TTTGGAAAGGACCAGCAAA-3′; reverse: 5′-CCTGCCATCTGTTTTCCA-3′; probe: 5′-6FAM-AAAGGTGAAGGGGCAGT AGTAATACA-Tamra-3′) as described in ref. 44 and RCAS amplification used primer/probe (forward: 5′-GTCAATAGAGAGAGGGGATGGA CAA A-3′; reverse: 5′-TCC ACA AGT GTA GCA GAG CCC-3′; probe: 5′-6FAM-TGG GTC

GGG TGG TCG TGC C-Tamra-3′) as described in ref. 45. Quantification was carried out on an ABI 7500FAST real-time thermocycler using an *in vitro* transcribed RNA standard. The assay has a limit of quantification of 10 copies per reaction, or 1 copy per μl of plasma.

**Bioluminescence imaging.** Bioluminescence imaging was performed with an IVIS Spectrum imaging system (Caliper Life Sciences). Mice were administered D-luciferin (Regis Technologies) at a dose of 150 mg kg$^{-1}$ intraperitoneally. Mice were anesthetized after receiving D-luciferin in a chamber with 3% isoflurane (Piramal Healthcare Limited) and placed on the imaging platform while being maintained on 2% isoflurane via a nose cone. Mice were imaged at 5 min post administration of D-luciferin using an exposure time of 5 s or longer to ensure that the signal acquired was within the effective detection range (above noise levels and below CCD saturation limit).

**Statistical analyses.** Statistical analyses were performed with Microsoft Excel and Prism 5.0 f (GraphPad). Data was compared using Student's *t*-test and one-way analysis of variance with Bonferroni corrections.

**Data availability.** All data are available within the article and its Supplementary Information file or from the authors upon request.

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

## Acknowledgements

This work was supported by a Takeda Pharmaceuticals New Frontier Science award and grants from NIAID/NIH; R01-AI050484 and R01-AI084860. We acknowledge assistance of the Penn Center for AIDS Research Immunology and Viral/Molecular Cores (NIH grant P30-AI045008). The VRC01 mAb was provided by Barney Graham of the Vaccine Research Center at the NIH.

## Author contributions

N.P., A.J.S., J.G., X.S., F.D., H.M., Y.Y., Y.K.T., B.L.M. and H.N. performed the experiments; N.P., G.A.D.-D., K.K., R.G.C. and D.W. designed the experiments; B.L.M., Y.K.T., T.D.M. and M.J.H. developed and prepared the lipid nanoparticles; N.P., A.J.S., Y.Y., F.S., R.G.C. and D.W. carried out analyses; N.P., K.K. and D.W. wrote the paper.

**Additional information**

**Competing financial interests:** Katalin Karikó and Drew Weissman are named on patents that describe the use of nucleoside-modified mRNA as a platform to deliver therapeutic proteins. The remaining authors declare no competing financial interests.

