## [Peer Review File · Nature Communications]

Reviewers' comments:

Reviewer #1, expert in vaccines and delivery systems (Remarks to the Author):

In this manuscript Pardi and colleagues describe a novel approach to express neutralizing anti-HIV antibodies in patients. Given the limitations of passive immunotherapy using recombinant antibodies, a number of studies have described gene therapy approaches using vectors like AAV or lentiviruses for long term expression of antibodies in patients. Notwithstanding, this approach is facing significant challenges mostly in terms of safety and limited duration due to emergence of neutralizing antibodies against the virus or antibody. The approach described in this study, the use of mRNA for transient expression of the encoded antibodies is novel, and offers potentially significant advantages over the gene therapy approaches, foremost addressing the safety concerns. While perhaps counterintuitive, direct injection of (modified and optimized) in vitro transcribed mRNAs in mice can be very effective in terms of quantity and duration of product synthesized as demonstrated by Weismann and colleagues in previous and this study. One limitation of the mRNA approach compared to the gene therapy approaches, perhaps a blessing in disguise, is the limited duration of antibody synthesis measured in days, up to a week. The "blessing" is that in the event of an emerging adverse reaction treatment can be readily stopped, or switching to alternative antibodies would more straightforward. It is also not unreasonable to expect that optimization of the protocol will improve the PK/PD of the approach. While may not be applicable to prophylactic vaccination of population at large, even at it's current state it appears to be ideally suited for instances where "cover" of limited duration with antibodies would suffice, such as in the setting of a "shock and kill" protocol to treat HIV latency.

From an experimental standpoint as a proof-of-concept study, it makes a compelling case.

Reviewer #2, expert in HIV therapies and delivery systems (Remarks to the Author):

The manuscript by Pardi and colleagues describes the use of mRNA-LNP for the delivery of broadly neutralizing antibody VRC01 and demonstrates that antibody delivered in this manner can protect a humanized mouse model of HIV infection.

Overall, the paper is clearly written and the authors present data from experiments that are well designed and well executed. However, the novelty of their findings are somewhat limited given the previously published work demonstrating the potential for VRC01 to protect humanized mice from HIV infection.

The studies presented raise several questions:

Where is the protein being expressed following IV administration of mRNA-LNP? Given that a luciferase transgene was used as a negative control, IVIS imaging of Luc control animals should be used to understand the tissues involved.

Can this approach work in immunocompetent animals? Would BALB/c mice given weekly mRNA-LNP produce a reasonable steady-state level of VRC01?

What is the immunogenicity against the VRC01 protein in BALB/c mice receiving weekly mRNA-LNP? The authors should demonstrate the concentration of mouse anti-human Fc antibody generated following weekly mRNA-LNPs in immunocompetent animals?

What was the % of hCD45 expressing cells that bearing CD3 or CD19? The methods indicate this was done but there are no graphs indicating the level of T or B lymphocyte engraftment in their humanized mice. Along these lines, do these mice have normal CD4 and CD8 ratios? What

happens to these ratios during infection?

What is the limit of detection of the viral load assay used in Figure 4? The plot implies 10 copies/uL but in this case, the true limit of detection is 10,000 copies/mL which is 10x less sensitive than what has been shown previously for humanized mice infected with HIV.

Overall, It is unclear what role the authors see for this approach in humans, or how it compares to simple passive transfer of antibody protein which has a well-established pipeline of production and a wealth of real-world experience. Greater discussion of this point would be welcome.

Answers for the comments

Thank you for considering our manuscript entitled “Administration of nucleoside-modified mRNA encoding broadly neutralizing antibody protects humanized mice from HIV-1 challenge”, NCOMMS-16-09902, for publication in Nature Communications. We greatly appreciate the time and effort of the editor and the 2 reviewers who critically read the manuscript. Please find how we addressed the comments from the editor and reviewers below.

Reviewer 1 did not ask for additional experiments or changes to the manuscript.

Reviewer 2

1. “Overall, the paper is clearly written and the authors present data from experiments that are well designed and well executed. However, the novelty of their findings are somewhat limited given the previously published work demonstrating the potential for VRC01 to protect humanized mice from HIV infection.”

In this study, VRC01 was used as a model to demonstrate the tremendous potentials of the mAb generated in vivo from injected nucleoside-modified mRNA and it is the first manuscript to describe this application. The novelty of these experiments is the therapeutic potential for in vivo synthesis of monoclonal antibody from encoding mRNA. We believe that delivery of monoclonal antibodies encoded as nucleoside-modified mRNA encapsulated in lipid nanoparticles is safer and more efficient than any other previously described delivery platforms (recombinant protein, viral, or DNA-based delivery tools) and can be applied to generate any antibody in vivo with extension to other extra and intracellular therapeutic proteins.

2. “Where is the protein being expressed following IV administration of mRNA-LNP? Given that a luciferase transgene was used as a negative control, IVIS imaging of Luc control animals should be used to understand the tissues involved”.

Protein expression from nucleoside-modified mRNA-LNPs was extensively studied in our previous work (please see reference #9). We have repeated the experiment demonstrating liver expression after intravenous delivery and included the data in the manuscript (please see page 6: lines 3-5 and Supplementary Figure 1).

3. “Can this approach work in immunocompetent animals? Would BALB/c mice given weekly mRNA-LNP produce a reasonable steady-state level of VRC01?” and “What is the immunogenicity against the VRC01 protein in BALB/c mice receiving weekly mRNA-LNP? The authors should demonstrate the concentration of mouse anti-human Fc antibody generated following weekly mRNA-LNPs in immunocompetent animals?”

Nucleoside-modified mRNA, while being non-immunogenic has no special properties that can make the encoded human protein (VRC01) non-immunogenic. Thus, an immune response to the human protein will definitely occur in immunocompetent (e.g. BALB/c) mice. We believe that repeated delivery of VRC01-encoding mRNA to BALB/c mice would not add new and useful information to the paper. We previously addressed the lack of immunogenicity of the nucleoside-modified mRNA and encoded protein when we delivered murine erythropoietin (muEPO) encoding mRNA to mice and found no antibody production against muEPO after 5 weekly injections (please see reference #17). We have repeated the experiment and included the data in the manuscript (please see page 8: lines 3-8 and Supplementary Figure 2).

4. “What was the % of hCD45 expressing cells that bearing CD3 or CD19? The methods indicate this was done but there are no graphs indicating the level of T or B lymphocyte engraftment in their humanized mice. Along these lines, do these mice have normal CD4 and CD8 ratios? What happens to these ratios during infection?”

We have included the CD3/CD19 data in the manuscript (please see page 8: lines 21-22 and Supplementary Figure 3b). CD4 and CD8 ratios were measured prior to HIV-1 challenge and have been included (please see page 8: lines 21-22 and Supplementary Figure 3c). We recognize that multiple methods can be used to determine infection in humanized mice. We chose to measure viral load in plasma and viral integration in spleen cells. Older methods, such as reversal of the ratio of CD4/CD8s, could be used, but we do not believe it adds any additional information.

5. “What is the limit of detection of the viral load assay used in Figure 4? The plot implies 10 copies/uL but in this case, the true limit of detection is 10,000 copies/mL which is 10x less sensitive than what has been shown previously for humanized mice infected with HIV.”

For each sample, the qRT-PCR reaction was run in duplicate on 5 µl RNA (equivalent to 10 µl plasma). The assay has a limit of quantification of 10 copies per reaction, or 1 copy per µl of plasma.

6. “Overall, It is unclear what role the authors see for this approach in humans, or how it compares to simple passive transfer of antibody protein which has a well-established pipeline of production and a wealth of real-world experience. Greater discussion of this point would be welcome.”

The manuscript demonstrates the superiority of VRC01 encoding nucleoside-modified mRNA-LNP delivery over recombinant purified VRC01 mAb delivery. 1.65 times higher plasma VRC01 protein levels were measured after the administration of 1/20th the amount of VRC01-encoding mRNA (please see page 9: lines 1-11 and

Figure 4a). We have discussed the advantages of using mAb encoding nucleoside-modified mRNAs (please see page 3: lines 15-21, page 4: lines 1-2, page 11: lines 9-18 and page 13: lines 9-22, page 14). The most important benefits are: (I) unlike recombinant mAb production that requires cell line production followed by complicated purification that differs for every protein, mRNA production is easy, fast, and uses identical methods for every coding sequence, (II) mRNA has the potential for cost-effective and highly scalable manufacturing, (III) small doses are sufficient to continuously express high levels of protein, (IV) the field of therapeutic proteins demonstrates many examples of adverse events due to misfolding or aberrant modification by cell line production, which are avoided with mRNA.

Reviewers' comments:

Reviewer #2 (Remarks to the Author):

While the authors have expanded the manuscript by including several new experiments, these data raise a few new questions that should be addressed.

Why is the level of VRC01 achieved so different in each experiment following administration of 30ug of mRNA? Fig 1 shows a peak of 160ug/mL, Fig 2 shows a steady state of 40ug/mL, Fig 4 shows 200ug/mL and Fig 5 shows 70ug/mL. Are there batch-to-batch variations in the production of the mRNA that influence activity in vivo? This should be addressed in the text.

Furthermore, the authors do not present evidence that the use of this approach for the delivery of antibodies is equally (or less) immunogenic than simple repeated protein injection. While the lack of immunogenicity against mouse EPO is clear, this misses the mark in that it masks any potential adjuvant effect of the mRNA that might be observed using an intentionally immunogenic protein. Repeated administration of mRNA encoding VRC01 should be compared to repeated administration of VRC01 protein in an immunocompetent animal to determine whether animals receiving VRC01-mRNA exhibit differences in immunogenicity as compared to purified protein. This is particularly relevant for the delivery of broadly neutralizing antibodies given their high level of somatic mutation and unusual CDR3 configurations that may be more immunogenic than typical antibodies.

One other concern regarding the humanized mouse experiments is that the protection experiments were ended only 2 weeks after challenge. Why were these experiments ended so soon? If these had been continued to later time points, would infection have been observed? Was infection simply delayed by the administration of the antibody or did they achieve sterilizing immunity by this approach?

Thank you for considering our manuscript entitled “Administration of nucleoside-modified mRNA encoding broadly neutralizing antibody protects humanized mice from HIV-1 challenge”, NCOMMS-16-09902A, for publication in Nature Communications. We greatly appreciate the time and effort of the editor and the reviewer who critically read the manuscript. Below, we detail our responses to the reviewer’s comments.

Reviewer 2

1. “Why is the level of VRC01 achieved so different in each experiment following administration of 30ug of mRNA? Fig 1 shows a peak of 160ug/mL, Fig 2 shows a steady state of 40ug/mL, Fig 4 shows 200ug/mL and Fig 5 shows 70ug/mL. Are there batch-to-batch variations in the production of the mRNA that influence activity in vivo? This should be addressed in the text.”

We completely agree, the high variability in plasma VRC01 antibody concentrations after injection of the same amount (30 µg) of VRC01 mRNA is confusing and needs to be clarified. We have not found significant batch-to-batch variability with nucleoside-modified mRNA-LNPs. Several reasons can explain the differences reported: (1) Four different types of mouse were injected with VRC01 mRNA, BALB/c, NSG, CD34-NSG humanized and BLT humanized. We used the same batch of VRC01 mRNA for the data in Figures 1, 2 and 5, where the main difference was mouse type (see page 18, lines 16-19). (2) The types of animals used in present study had different weights that resulted in varying VRC01 mRNA doses (from 1 to 1.4 mg/kg). We

originally planned to use the same dose (μgs) for all treatments to be consistent, but in hindsight, we should have dosed as mg/kg.

We inserted mg/kg as the unit of mRNA doses in the text to clarify this concern. In Fig. 1, BALB/c mice were used and the dose was 1.4 mg/kg (please see page 2, line 8; page 6, line 7; page 13, lines 10-11; and Fig. 1 legend). In Fig. 2, NSG mice were weekly injected with a dose of 1 mg/kg of VRC01 mRNA, blood was collected prior to each injection, so the 40 $\mu\text{g/ml}$ VRC01 concentration was measured 7 days after injection (in all other studies VRC01 levels were measured 24 hours post-injection) (please see page 2, line 10; page 6, line 20; and Fig. 2 legend). We have described the timing of the measurement more precisely on page 6, line 21. In Fig. 4, BLT mice were injected with 1.4, 0.7 or 0.35 mg/kg doses (please see page 9, lines 1-2; page 14, line 3; and Fig. 4 legend). In Fig. 5, CD34-NSG mice were treated with 1 mg/kg of VRC01 mRNA (please see page 10, line 1; and Fig. 5 legend).

2. “Furthermore, the authors do not present evidence that the use of this approach for the delivery of antibodies is equally (or less) immunogenic than simple repeated protein injection. While the lack of immunogenicity against mouse EPO is clear, this misses the mark in that it masks any potential adjuvant effect of the mRNA that might be observed using an intentionally immunogenic protein. Repeated administration of mRNA encoding VRC01 should be compared to repeated administration of VRC01 protein in an immunocompetent animal to determine whether animals receiving VRC01-mRNA exhibit differences in immunogenicity as compared to purified protein. This is particularly

relevant for the delivery of broadly neutralizing antibodies given their high level of somatic mutation and unusual CDR3 configurations that may be more immunogenic than typical antibodies.”

We do not think it would be possible to measure slight differences in the immunogenicity of a highly immunogenic protein comparing mRNA to protein delivery, as a measure of potential RNA adjuvant activity, which we demonstrated does not exist. A negative result will not be useful to the field. We know that with the second delivery using either method, the protein is rapidly cleared due to the immune response developed. We added the EPO data (Supplementary Fig. 2) that demonstrated the absence of immune response induction in mice after repeated delivery of mRNA encoding a murine protein. As we understand, this comment questioned whether the mRNA can act as an adjuvant and make the encoded “host” protein immunogenic, as has been observed when the host’s EPO protein was generated from an injected viral vector (AAV) or produced in vitro and delivered as a protein. We feel that our argument is significant and strong when using the identical protein, murine EPO, delivered using nucleoside-modified mRNA, and demonstrating the absence of any immune response.

We think the question we are now asked to answer is whether a murine monoclonal that could contain an immunogenic idotype would induce different amounts of immunogenicity when delivered as a protein versus mRNA. Similar questions have been raised with the AAV delivery of monoclonals and other proteins, which still have not been answered due to the complexity of the required experiments. These

experiments are beyond the scope of this manuscript, which is the first description of a therapeutic protective monoclonal delivered using the encoding mRNA.

3. “One other concern regarding the humanized mouse experiments is that the protection experiments were ended only 2 weeks after challenge. Why were these experiments ended so soon? ”

We could have used the alternative method and allowed the infections to proceed for many weeks to months to determine whether infection was completely blocked, as suggested by the reviewer. We chose the short time frame, as our experience with the mice produced by our Xenograft Facility is that they quickly develop graft versus host disease resulting in the need to euthanize. We felt that we would have the greatest ability to measure our endpoints by using early termination and measurement of splenic HIV DNA. We found when measuring HIV DNA in the spleen, the major site of residence of human T cells, that sensitivity was lost with increasing time post-infection due to virus-mediated CD4⁺ T cell depletion. Thus, we analyzed early time points post-challenge, to determine if mAb-treated mice were protected from infection.

“If these had been continued to later time points, would infection have been observed? Was infection simply delayed by the administration of the antibody or did they achieve sterilizing immunity by this approach?”

Based on our observation that VRC01 mRNA-injected mice had undetectable HIV gag DNA in the spleen (PCR assay sensitivity of 3 copies), we believe that the level of protection in these challenged mice was sterilizing, and delayed virus replication is extremely unlikely.